# Alterations in Herbage Yield, Antioxidant Activities, Phytochemical Contents, and Bioactive Compounds of Sabah Snake Grass (*Clinacanthus Nutans* L.) with Regards to Harvesting Age and Harvesting Frequency

**DOI:** 10.3390/molecules25122833

**Published:** 2020-06-19

**Authors:** Nur Mardhiati Afifa Abd Samat, Syahida Ahmad, Yahya Awang, Ros Azrinawati Hana Bakar, Mansor Hakiman

**Affiliations:** 1Department of Crop Science, Faculty of Agriculture, Universiti Putra Malaysia, Serdang 43400 UPM, Selangor, Malaysia; gs54768@student.upm.edu.my (N.M.A.A.S.); yahya_awg@upm.edu.my (Y.A.); roshana9090@gmail.com (R.A.H.B.); 2Department of Biochemistry, Faculty of Biotechnology and Biomolecular Sciences, Universiti Putra Malaysia, Serdang 43400 UPM, Selangor, Malaysia; syahida@upm.edu.my; 3Laboratory of Sustainable Resources Management, Institute of Tropical Forestry and Forest Product, Universiti Putra Malaysia, Serdang 43400 UPM, Selangor, Malaysia

**Keywords:** herbage yield, antioxidant, flavonoid, phenolic, c-glycosyl flavone, harvesting age, harvesting frequency

## Abstract

Sabah snake grass or *Clinacanthus nutans* has drawn public interest having significant economic benefits attributable to the presence of phytochemicals and several interesting bioactive constituents that may differ according to harvesting age and harvesting frequency. The current study was aimed to evaluate the effect of harvesting age and harvesting frequency towards herbal yield, antioxidant activities, phytochemicals synthesis, and bioactive compounds of *C. nutans*. A factorial randomized completely block design with five replications was used to illustrate the relationship between herbal yield, DPPH (2, 2-diphenyl-1-picrylhydrazyl) and ferric reducing antioxidant power (FRAP) assays, total phenolic and flavonoid content affected by harvesting age (week 8, 12, and 16 after transplanting), and harvesting frequency (harvest 1, 2, and 3). The bioactive compounds by HPLC were also determined to describe the interaction effect between both harvesting age and harvesting frequency. The yield, antioxidant activities, and phytochemical contents were gradually increased as the plant grew, with the highest recorded during week 16. However, the synthesis and activities of phytochemicals were reduced in subsequent harvests despite the increment of the herbal yield. All bioactive compounds were found to be influenced insignificantly and significantly by harvesting age and harvesting frequency, respectively, specifically to shaftoside, iso-orientin, and orientin. Among all constituents, shaftoside was the main compound at various harvesting ages and harvesting frequencies. These results indicated that harvesting at week 16 with 1st harvest frequency might enhance the yield while sustaining the high synthesis of polyphenols and antioxidant activities of *C. nutans*.

## 1. Introduction

A wide number of studies in current years has been focusing on the secondary metabolites due to the nature of antioxidant that potentially modulates human metabolism. These metabolites, which include saponins, tannins, anthocyanins, lignin, phenolics, and flavonoids, have been practically used by more than 80% of world populations to prevent and treat various health problems, such as cancer, diabetes, and high blood pressure [1]. Flavonoids, as the largest family having a low molecular structure of phenolic secondary metabolites, are well known to be associated with great length of pharmacological activities, which widely spread throughout the natural plants, including most the medicinal plants [2]. These compounds are receiving a number of considerable attention due to documented protective role against many diseases, which may be attributed to the biological effects of antioxidant activity against reactive oxygen species [3]. Due to the presence of this compound, medicinal plants have been utilized extensively in the world of pharmaceuticals as an alternative regiment against various diseases.

Native to tropical Asia countries, *Clinacanthus nutans* or commonly known as Sabah snake grass is well-reputed in Thai folklore medicine due to its ability to cure various ailments, such as skin inflammation and snake bites. For the past few decades, the research for *C. nutans* has been progressively done in a rapid trend. Broad-spectrum of pharmacological activities, which includes antioxidant [4], antiviral (herpes simplex virus (HSV), varicella-zoster virus, papillomavirus, and dengue virus) [5], anti-inflammatory [6], antibacterial [7,8,9], analgesic [10,11,12], antivenom [13,14,15], immunomodulating [16,17], neuroprotective [18,19,20], antidiabetic [21,22,23], α-glucosidase inhibitory [24], lipid-elevated inhibition [25], plasmid DNA protective [26], anti-angiogenic [27], anticancer [28], anti-proliferative [28], and antitumor as the latest biological action identified [29], has been very well reported in previous studies. Owing to significant economic benefits towards human health, the demand for this herb has been increased in recent years.

Although these constituents offer many substantial effects in human welfares, the biosynthesis of polyphenolic compounds attributed the biological activities may often modulate by various factors, which include cultivars, genotypes, environmental conditions, such as light intensity, relative humidity, and cultivation practices. Harvesting application as part of the cultivation practices has been known to have a significant role in influencing the quantitative and qualitative aspects in plants, resulting in an increment in growth, yield, and value of crops [30,31,32,33,34,35]. It is indeed shown that age-related factor has effectively enhanced the quantity and quality of the plant, specifically in phytochemical content and antioxidant activities in several crops [36,37,38,39]. However, repetitive and continuous harvesting has led to the degradation of these compounds [40,41,42]. These reported studies have essentially addressed to the agricultural and operational process done for *C. nutans* in Malaysia as farmers often repeatedly ratoon the plants for better yield increment, overlooking the effect of the practices on the quality aspect of *C. nutans* itself. Studies covering the response of phytochemical synthesis and bioactive compounds remain deficient despite the numerous findings regarding its beneficial aspects. As valuable natural resources, it is desirable to undertake further studies, gathering relevant information through harvesting management as alternative ways. Since phytochemicals’ content vary according to age and continuous harvesting, it is necessary to harvest the medicinal plants of a particular age in particular harvesting frequencies to sustain the medicinal potency of the plant. Therefore, this current study was aimed to evaluate the influence of interaction between harvesting age and harvesting frequencies towards the plant’s performances in synthesizing phytochemicals as well as bioactive constituents of *C. nutans*.

## 2. Results and Discussions

### 2.1. Herbal Yield

Both harvesting age and harvesting frequency significantly interacted with each other on the herbal production (Table 1, Appendix A). The highest production was recorded at week 16, while the lowest recorded at week 8. Similar results were obtained at different levels of harvesting frequency, where the plants that were harvested repeatedly via cutting produced more herbal yield.

Under age-related harvesting, the manifested yield can be driven by many factors as plants constantly experience various stresses throughout the growth period [43]. These stresses lead to growth alteration in the shape and size of the plants to be well-adapted to environmental changes. It was presumed that light acquirement for photosynthesis has forced the assimilates produced by the leaves to be accumulated in the developing sink (new leaves), consequently leading to high herbal yield [44]. The mechanism may equally apply to the harvesting frequency. According to Marasha et al. [42], cutting operations in frequent harvesting may enhance the vegetative growth and yield of crops. A limitation of resource supply and changes in plant size via complete detachment of the aboveground part of plants possibly led the plant to shift its carbohydrates reservation to the leaf (dominant sink) for light acquisition, clarifying the increment of herbal yield at various harvesting frequencies (Table 1).

### 2.2. Total Phenolic Content (TPC) and Total Flavonoid Contents (TFC)

As shown in Table 1, TFC and TPC of *C. nutans* were significantly influenced by harvesting age and harvesting frequency (*p* ≤ 0.05, Appendix B). From the table shown, the concentration of TPC, which ranged between 5.82 and 10.84 mg GAE/g DW, was much slightly lower than the concentration of TFC, which ranged between 4.31 and 7.32 mg QE/g DW for both treatments. These results were in line with a number of studies signifying that flavonoid is a part of the phenolic compounds, making the quantity to be generally lower compared to phenolic contents. Under age-related, both TPC and TFC of *C. nutans* were responded proportionally with time (*p* ≤ 0.01, respectively) with the highest TPC and TFC recorded during week 16 of harvesting (average value: 10.84 mg GAE/g DW and 7.32 mg QE/g DW, respectively), statistically followed by week 12 (average value: 6.87 mg GAE/g DW and 6.21 mg QE/g DW, respectively) and week 8 (average value: 6.17 mg GAE/g DW and 4.31 mg QE/g DW, respectively) containing the lowest value. The production of TFC and TPC of Sabah snake grass, however, degraded following the number of harvesting frequency in a linear manner (*p* ≤ 0.01, respectively), with the lowest concentrations recorded during third harvesting frequency (average value: 5.82 mg GAE/g DW and 4.69 mg QE/g DW, respectively), followed by second (average value: 7.75 mg GAE/g DW and 6.02 mg QE/g DW, respectively) and first harvesting frequency (average value: 10.30 mg GAE/g DW and 7.13 mg QE/g DW, respectively). Although no interaction was analyzed for TFC between both treatments, the production of both TFC and TPC in *C. nutans* shared a similar pattern where both polyphenolics were enhanced significantly as the plant grew but substantially reduced as the plant was repeatedly harvested.

In a different study on *C. nutans* itself, similar results were also obtained by Ghasemzadeh et al. [37], who confirmed that the production of phytochemicals remained increased until the age of 6 months, which then decreased until the age of 1 year. Sharing the same family to *C. nutans*, *Andrographis paniculata* (Hempedu bumi) also displays an increment in the content of andrographolide until a certain point of harvesting stage and then decreases afterward due to the senescence of leaves during reproductive phase [33]. Current results on reduction in phytochemicals at different harvesting frequencies were in agreement with Toit [45] and Hue [41], stating the limitation of substrates for synthesizing the secondary metabolites as the probable cause. Marasha [42] explained that repetitive harvesting might help to stimulate and improve the vegetative growth of the plant. However, rapid regrowth of plant following the above-ground removal would require a high amount of carbohydrates acquisition, which restricted the substrate for carbon-based secondary metabolites (CBSM), explaining the degradation of phytochemicals at varying levels of harvesting frequency. Consistency of previous findings and current results justified that the phenolic and flavonoid content of *C. nutans* could highly be influenced by harvesting age and harvesting frequency.

### 2.3. Radical Scavenging Activity (DPPH) and Ferric Reducing Power Assay (FRAP)

In vitro, the specific antioxidant activity can be analyzed by different methods, yielding information on the effectiveness of phenolics to neutralize free radicals. Among in vitro assays, the DPPH-based technique is, furthermore, quick, simple, relatively straightforward to perform, and inexpensive compared to other assays. However, antioxidant activity should not be deduced depending on a single antioxidant test model. In this experiment, samples were subjected to both DPPH (2, 2-diphenyl-1-picrylhydrazyl) and ferric reducing antioxidant power (FRAP) assays to estimate the capacity of *C. nutans* in scavenging the free radicals and measure the comparison data obtained between these assays.

Pearson correlation analysis in Figure 1 demonstrated a highly significant and positive correlation in linear form between the DPPH and FRAP results. This finding was undoubtedly in line with a number of previous studies [46,47,48,49,50,51], considering these two assays displayed a complementary mechanistic basis [52], and supposedly believing that conducting two or more assays could enhance the overall evaluation of the antioxidant activity of plant extracts as each assay demonstrated various antioxidant performances [53]. Although these assays vary in mode of actions, both assays considerably exhibit a high degree of redundancy on account of sharing akin mechanism. As such, the selection of an assay before running the analysis is important as each assay has its own pros and cons. Compared to DPPH, FRAP is simple, rapid, and inexpensive [54], yet issues, such as color interferences in extracts and slow development of color, have been reported upon utilizing FRAP as an antioxidant assay [55,56,57]. ABTS (2,2′-azino-bis(3-ethylbenzothiazoline-6-sulfonic acid) and Oxygen Radical Absorbance Capacity (ORAC) assays should be in favor of future studies to analyze the antioxidant action in *C. nutans*.

The DPPH scavenging activity and FRAP activity of *C. nutans* at different levels of harvesting age and harvesting frequency are presented in Table 2. Based on Table 2, the DPPH activity varied significantly and independently to the respective harvesting age and harvesting frequency (*p* < 0.05, Appendix B). In comparison to DPPH, FRAP activity exhibited a substantial effect and interaction effect at varying levels of harvesting age and harvesting frequency. Both DPPH and FRAP activities displayed a similar trend, where both of these antioxidant activities were increased simultaneously with harvesting age, with the highest DPPH and FRAP value observed at week 16, followed statistically by week 12 and week 8. Similar to phytochemicals, both DPPH and FRAP activities were declined with a number of harvesting frequency, having the lowest value recorded during third harvesting, while the first harvesting frequency contained the highest antioxidant activities. Although the trend exerted by these antioxidant activities was similar, the changes in value could be seen in the FRAP assay compared to DPPH. In addition, the pattern shown by FRAP activity was in line with phenolic contents, suggesting that FRAP assay favored the phenolic compounds since both phenolic and FRAP activities were interactively and substantially affected by harvesting age and harvesting frequencies. The occurrence of antioxidant activities in *C. nutans* has been very well affirmed by many studies [58,59,60,61]. Pannagtech [61] noted that *C. nutans* had the capability of inhibiting the oxidative stress in the 2,2′-azobis (amidinopropane) dihydrochloride (AAPH)-induced red blood cell lysis, which was attributed to the exertion of free radical scavenging activity and ferric reducing antioxidant power in plants.

### 2.4. Bioactive Compounds (C-Glycosyl Flavone)

All known compounds displayed a good constancy with the corresponding standard solution in terms of retention times, confirming the validity of the results (Figure 2). The C-glycosyl flavone compounds at different levels of harvesting age and harvesting frequency are shown in Table 3. Briefly, current results presented below revealed that the shaftoside was the main flavone compound as it was constantly detected at all levels of both treatments, ranging from 5.32 µg/g to 7.39 µg/g for harvesting age and 11.27 µg/g to 3.20 µg/g for harvesting frequency. The remaining identified flavone compounds were detected in small amount, ranging from highest recorded in iso-orientin (harvesting age: 1.59–2.31 µg/g, harvesting frequency: 0.49–3.07 µg/g), followed statistically by orientin (harvesting age: 0.56–0.66 µg/g, harvesting frequency: 0.23–1.09 µg/g), iso-vitexin (harvesting age: 0.45–0.65 µg/g, harvesting frequency: 0.36–0.83 µg/g), and vitexin as the lowest amount presented (harvesting age: 0.09–0.32 µg/g, harvesting frequency: 0.23–0.51 µg/g). Although all compounds were present at most levels of treatments, a significant variation could be observed only in three compounds, namely, shaftoside, iso-orientin, and orientin, specifically for harvesting frequency (*p* < 0.05, Table 3, Appendix B). An abundant amount of shaftoside, in the current study, was consistent with the results of Chelyn et al. [62], where the largest amount of shaftoside was found in all locations, namely, Taiping, Perak, Kota Tinggi, Johor and Sendayan, and Negeri Sembilan. Stalikas [63] explained that the C-glycosyl flavone was favorable for the chemical markers due to the wide range of structural diversity. Since shaftoside was found to be abundant at all levels of harvesting age and harvesting frequency, it was suggested that shaftoside would be a possible chemical marker for *C. nutans* raw materials.

### 2.5. The Correlation between Herbal Yield, Antioxidant Activities, and Identified Polyphenolic Compounds

To date, it is acknowledged that phytochemicals are the major contributors to the activity of antioxidants in most potent medicinal and aromatic plants. Since there was a distinct demonstration in both treatments in this study, correlation data were divided into two tables. As can be seen in Table 4, for harvesting age, the herbal yield of *C. nutans* was significantly and positively correlated with phytochemicals and antioxidants. Under frequency-related harvesting, however, the yield showed a negative relationship with phytochemicals and antioxidants (Table 5). Results also showed that the DPPH activity was significantly correlated with FRAP activity in both harvesting age and harvesting frequency. The same results were also applied to total phenolics and total flavonoids, where both of these phytochemical contents were correlated significantly with both DPPH activity and FRAP activity.

In relation to the C-glycosyl flavone, a significant correlation between these constituents, phytochemicals, and antioxidant activities was only noticeable in the harvesting frequency and not in than harvesting age. There were three known compounds that demonstrated a significant correlation between phytochemicals and antioxidant activities, namely, shaftoside, iso-orientin, and orientin (Table 5). These correlations indeed elucidated the exhibition of antioxidants in *C. nutans,* which were greatly due to these identified compounds. Although the literature lacks the correlation between these compounds with antioxidant capacity, it is agreeable that phenylpropanoid glycosides possess high antioxidant power [64]. The phenylpropanoid pathway is an important biosynthesis of secondary metabolites in medicinal plants derived from phenylalanine and tyrosine [65]. A common metabolic fate, glycosylation, is a process that is known to regulate flavonoid’s structural stability [66]. Glycosylation is not only highly correlated with plant taxonomy [67], it is also recognized as the major regulator of phenylpropanoid’s availability, stability, toxic potential, and biological activity in plants [68]. The uncorrelated data between antioxidants and these constituents were shown at different levels of harvesting age, feasibly due to several factors. Seeing that this study was undertaken in an open field, various changes in exogenous sources, such as light, temperatures, soil nutrition, and textures, occurring throughout the plant growth process, presumably altered or disrupted the synthesis and structure of bioactive compounds [68]. Managing both negative and positive interactions with the environment, the plants constantly synthesize intricate and dynamic mixtures of secondary metabolites to counter explicit environmental conditions [69,70,71,72].

The relationship, reported between phytochemicals and antioxidant activities, is similar in several studies [37,73,74]. Flavonoids, as part of the phytochemicals, act as antioxidants, contributing to most of the biological activities in higher plants, including medicinal plants. Phytochemicals tend to respond simultaneously and linearly with antioxidants by direct correlation, indicating the responsibility of phytochemicals for the antioxidant activities exerted in plants [75,76,77]. This will indirectly affect the known individual of polyphenolic compounds, which can be seen in kaempferol and quercetin, while other constituents show poor correlation results [37]. The consistency of current results with the previous findings implied that the identified constituents, known as shaftoside, iso-orientin, and orientin flavone compound, could be the key, determining the antioxidant activities in *C. nutans* at different levels of harvesting frequency.

## 3. Materials and Methods

### 3.1. Standards and Chemicals

All chemicals were analytical and HPLC (HPLC analysis)-reagent grade. The chemicals included methanol (Systerm ®, ChemAR, Shah Alam, Malaysia), Folin-Ciocalteau reagent (Sigma-Aldrich, St. Louis, MO, USA), sodium carbonate (Sigma-Aldrich, St. Louis, MO, USA), gallic acid (Sigma-Aldrich, St. Louis, MO, USA), sodium nitrite (Systerm ®, ChemAR, Shah Alam, Malaysia), aluminium chloride (Systerm ®, ChemAR, Shah Alam, Malaysia), sodium hydroxide (Systerm ®, ChemAR, Shah Alam, Malaysia), quercetin (Sigma-Aldrich, St. Louis, MO, USA), 2,2-diphenyl-1-picrylhydrazyl (DPPH) (Sigma-Aldrich, St. Louis, MO, USA), sodium acetate buffer (Sigma-Aldrich, St. Louis, MO, USA), 2,4,6-tri [2-pyridyl]-s-triazine (TPTZ) (Sigma-Aldrich, St. Louis, MO, USA), iron (III) chloride (Systerm ®, ChemPur, Shah Alam, Malaysia), hydrochloric acid (Sigma-Aldrich, St. Louis, MO, USA), ferrous sulphate (Sigma-Aldrich, St. Louis, MO, USA), methanol HPLC grade (Sigma Aldrich, St. Louis, MO, USA), shaftoside (Sigma-Aldrich, St. Louis, MO, USA), orientin (HPLC grade, Sigma-Aldrich, St. Louis, MO, USA), iso-orientin (HPLC, Sigma-Aldrich, St. Louis, MO, USA), vitexin (Sigma-Aldrich, St. Louis, MO, USA), isovitexin (HPLC grade, Sigma-Aldrich, St. Louis, MO, USA), Milli Q water, distilled water, orthophosphoric acid (HPLC grade, Sigma-Aldrich, St. Louis, MO, USA), acetonitrile (HPLC grade, Sigma-Aldrich, St. Louis, MO, USA).

### 3.2. Planting Materials, Experimental Field, and Treatments

This study was carried out in open field conditions at Kompleks Ladang Bersepadu, Ladang 10, Universiti Putra Malaysia (UPM), Serdang, Selangor, Malaysia. The soil of the experimental area was clay loamy in texture, slightly acidic (pH 5.82), with good cation exchange capacity (8.21 cmol/kg). The soil nutrients were as follows: C (1.8%), N (0.12%), P (18.48 µg/g), K (141.5 µg/g), Ca (705.1 µg/g), and Mg (83.4 µg/g). The plants were propagated through stem cuttings and prepared in polyethylene bags filled with topsoil. The stems were then arranged and left aside under shading areas, allowing them to grow for a month until they grew evenly with daily irrigation. When propagated stems reached 90% uniformity in height, the stems were transplanted to the prepared field plots arranged by factorial randomized complete block design with five replications. The plants were harvested according to the treatments consisting of three levels of harvesting age (weeks 8, 12, and 16) and harvesting frequency (harvest 1, 2, and 3). Leaves were separated from the stems and oven-dried at 35 °C. Once dried, the samples were weighed, blended into powdered form, and stored at −20 °C for further analysis.

### 3.3. Extraction Method

The 0.5 g of every ground sample was extracted with 15 mL aqueous methanol (methanol: water, 80–20 *v*/*v*) for 3 h at room temperature in an orbital shaker. The extracts were separated from the residues by filtering through Whatman No.1 filter paper. The extracts were then transferred into airtight amber vials and stored at −80 °C until used for analysis.

### 3.4. Determination of Total Phenolic Content (TPC)

The content of phenolics, from the extract of *C. nutans,* was evaluated by using the Folin–Ciocalteau reagent-based assay, according to Barku [74] with slight modification. A 5 mL of Folin–Ciocalteau reagent was added to the 200 µL of samples, and the solutions were allowed to stand for 10 min at 25 °C. Later, 4 mL of sodium carbonate was added into the solutions, and the solutions were kept in total darkness for 20 min at room temperature. The absorbance reading of the blue color mixture was measured at 765 nm using an ultraviolet-visible (UV) spectrophotometer (Shimadzu, Kyoto, Japan). Gallic acid was used as a standard for the calibration curve, and the samples were expressed as mg GAE/g plant dry weight.

### 3.5. Determination of Total Flavonoid Content (TFC)

The flavonoid content of *C. nutans* was evaluated according to aluminum chloride assay [37]. The 1 mL of each extract was transferred into 10-mL volumetric flasks containing 4 mL of 80% methanol. Then, 0.3 mL of sodium nitrite (NaNO_2_) solution (1:5, *w*/*v*) was added to the flasks and allowed to stand for 6 min at room temperature. Next, 0.3 mL of aluminum chloride (AlCl_3_) solution (1:10, *w*/*v*) was added into the mixtures and kept for 6 min at room temperature. Finally, 2 mL of sodium hydroxide (NaOH) solution (1 M) was mixed and kept for 10 min at room temperature. The absorbance reading of yellow mixtures was measured at 510 nm by ultraviolet-visible (UV) spectrophotometer (Shimadzu, Kyoto, Japan). Quercetin was used as a standard for the calibration curve, and the samples were expressed as mg quercetin/g plant dry weight.

### 3.6. DPPH Radical Scavenging Activity Assay

The antioxidant activity of the aqueous methanol extract of *C. nutans* was evaluated using 2,2-diphenyl-1-picrylhydrazyl (DPPH) radical, based on the electron transfer reaction between DPPH reagent and the plant extract according to the method [77]. A 0.2 mM solution of DPPH in pure methanol was prepared, and 2 mL of this solution was added to 2 mL of all extracts at various concentrations. The mixtures were shaken gently and allowed to stand for 30 min at room temperature. The absorbance for both positive control (DPPH solution) and samples was measured at 517 nm against methanol as a blank using an ultraviolet-visible (UV) spectrophotometer (Shimadzu, Kyoto, Japan). The inhibition percentage of the absorbance was calculated as follows:Inhibition % = [(absorbance of control - absorbance of sample)/absorbance of control)] × 100

### 3.7. Ferric Reducing Antioxidant Power Assay (FRAP)

The ability to reduce ferric ions of the aqueous methanol extract was evaluated using the method described by Benzie and Strain [78]. The working FRAP reagent with the ratio of 10:1:1 was developed daily from 300 mM sodium acetate buffer, 10 mM 2,4,6-tri [2-pyridyl]-s-triazine (TPTZ) in 40 mM hydrochloric acid (37%), and 20 mM FeCl_3_. The pH value of the buffer was checked and maintained at pH 3.6. A ferrous sulfate (FeSO_4_·7H_2_O) was used as a standard, and a linear curve was prepared for various concentrations. For the analysis, 200 µL of the extract was added to 3 mL freshly prepared FRAP solution, and the reaction mixture was incubated for 30 min at 37 °C in the water bath. Then, the absorbance of the sample readings was recorded at 593 nm using an ultraviolet-visible (UV) spectrophotometer (Shimadzu, Kyoto, Japan). The solvent without sample was taken as a blank, and the FRAP value was calculated from the differences in the absorbance of a sample and a blank. The FRAP value based on the ability to reduce ferric ions of the extract was expressed as µM(Fe(II)/g dry mass.

### 3.8. Separation and Analysis of Bioactive Compounds (C-Glycosyl Flavone) by HPLC

The determination of the bioactive compounds of *C. nutans* was done, based on Chelyn [62], with slight modifications. The methanolic extract of *C. nutans* was prepared by adding methanol 80% (10 mL) to 0.5 g of powdered leaves. The mixtures were then vortexed for 10 min to dissolve the samples and sonicated for 30 min at 45 °C in an ultrasonic bath. Then, the samples were filtered using Whatman no. 1 filter paper, and the collected supernatant was subjected to HPLC analysis. The standard stock solutions of shaftoside, orientin, iso-orientin, vitexin, and iso-vitexin were prepared in HPLC grade methanol and stored at 4 °C. HPLC analysis was performed on Waters Alliance (E2695 and 2998 PAD Detector). The chromatographic separation was performed using C18 (250 × 4.6 mm, 5 µm). The solvent system consisted of mixtures of Milli Q water with orthophosphoric acid at pH 2.5 (solvent A) and absolute acetonitrile HPLC grade (solvent B). The solvents were degassed before delivering into the system. Samples for HPLC analysis were filtered through a 0.45 µm membrane filter. The flow rate and the injection volume were 0.8 mL/min and 20 µL, respectively. The signal was monitored at 280 nm. The gradient HPLC condition was as follows: 0–10 min 5–10% B; 10–40 min 10–40% B; 40–48 min 40–100% B; 48–55 min 100% B; 55–58 min 100–5% B; 58–64 min 5% B. A linear relationship between absorbance and concentration of standard was prepared based on the Beer-Lambert law. The quantification of compounds in samples was done based on a linear regression equation of standard, Y = aX ± b, where X was the concentration of phenolic, and Y was the peak height of phenolic obtained from HPLC. HPLC chromatogram for all standard solutions with retention time is shown in Figure 2.

### 3.9. Statistical Analysis

Collected data from the study were presented as mean ± SD of five replications, and means were analyzed with the analysis of variance (ANOVA) by statistical analysis system (SAS, Ver. 9.3, Cary, NC, USA). Means were compared using Duncan multiple range test (DMRT) at *p* < 0.05.

## 4. Conclusions

Generally, the results obtained from this study indicated that the yield, phytochemicals, antioxidant activities, and bioactive constituents of *Clinacanthus nutans* were changing according to varying levels of harvesting age and harvesting intervals. Plant harvested at week 16 during first harvesting frequency certainly produced the highest phenolic, flavonoids, and antioxidant activities, and then decreased with repeating harvesting. Of all known constituents, the shaftoside flavone compound was suggested as a chemical marker because it was the major compound found at all levels of harvesting age and harvesting frequency. *C. nutans* was considered to be a valuable resource, and it exhibited a good natural antioxidant.

However, the present scientific study was quite insufficient, preliminary, and fundamentally oriented. More sophisticated evaluation and pathway analyses with other biological and therapeutic effects due to biotic and abiotic factors should be highlighted for the next experiment. It would also be beneficial to have more experimental studies that could describe the correlation of the isolated phytochemicals from *C. nutans* using multiple and latest software analysis.

Regardless, the information obtained from this study would be greatly beneficial not only for the researchers but also for the farmers who are interested in cultivating the plants. It should be highlighted that continuous cutting of plants has a substantial negative effect on the quality despite the increment on the yield. Quality and quantity aspects should be in equilibrium balance when it comes to operating agricultural practices of any crop in order to produce high-quality plant-based products.

## Figures and Tables

**Figure 1 molecules-25-02833-f001:**
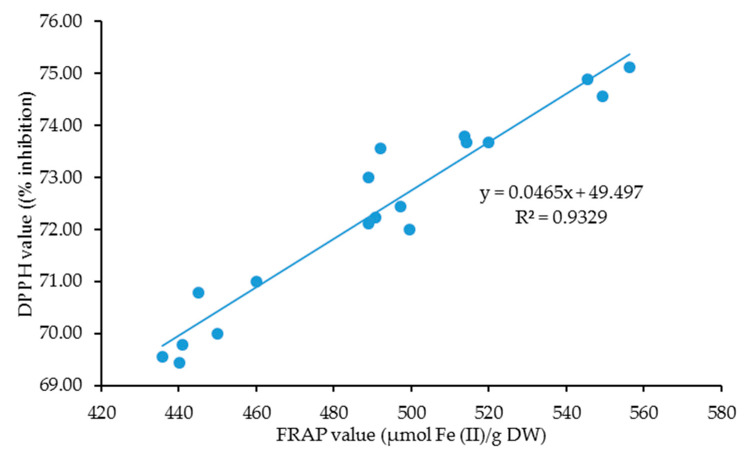
The correlation between DPPH value (% inhibition) and FRAP value (µmol Fe (II)/g DW) for a combination of both harvesting age and harvesting frequency.

**Figure 2 molecules-25-02833-f002:**
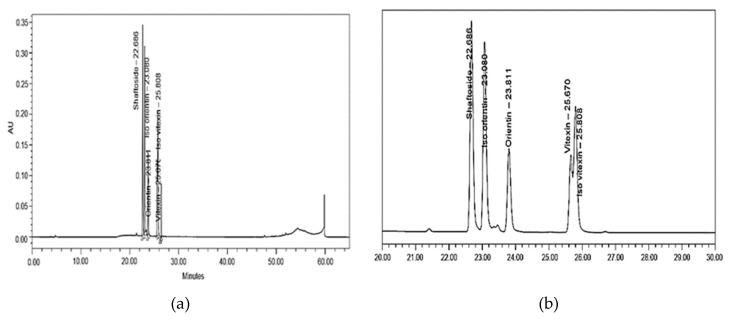
HPLC chromatogram of the standard solution in (**a**) full and (**b**) zoom. Shaftoside (RT: 22.686), iso-orientin (RT: 23.080), orientin (RT: 23.811), vitexin (RT: 25.670), and iso-vitexin (RT: 25.808). RT refers to retention time.

**Table 1 molecules-25-02833-t001:** Herbal yield, total phenolic, and total flavonoid content of the methanolic extracts of *C. nutans* at varying levels of harvesting age and harvesting frequency.

Treatment	Herbal Yield (g)	Total Phenolic Content (mg GAE/DW)	Total Flavonoid Content (mg quercetin/g DW)
Harvesting age (HA)			
Week 8	15.43 ± 1.51 ^c^	6.17 ± 0.17 ^c^	4.31 ± 0.31 ^c^
Week 12	29.00 ± 1.79 ^b^	6.87 ± 0.11 ^b^	6.21 ± 0.33 ^b^
Week 16	129.56 ± 2.41 ^a^	10.84 ± 0.24 ^a^	7.32 ± 0.40 ^a^
Harvesting frequency (HF)			
Harvest 1	17.07 ± 1.26 ^c^	10.3 ± 0.14 ^a^	7.13 ± 0.38 ^a^
Harvest 2	61.08 ± 2.21 ^b^	7.75 ± 0.19 ^b^	6.02 ± 0.33 ^b^
Harvest 3	96.04 ± 2.23 ^a^	5.82 ± 0.19 ^c^	4.69 ± 0.34 ^c^
F test			
Harvesting age	**	**	**
Harvesting frequency	**	**	**
HA × HF	**	**	ns

Data are means of five replication measurements ± standard deviation. Means with different letters (a, b, c) at different levels of harvesting age and harvesting frequency were significantly different at *p* < 0.05 based on Duncan Multiple Range Test (DMRT) mean separation. DW: dry weight. ** is significant at *p* < 0.01. ‘ns’ is not significant.

**Table 2 molecules-25-02833-t002:** DPPH and FRAP activities of the methanolic extracts of *C. nutans* at varying levels of harvesting age and harvesting frequency.

Treatment	DPPH Scavenging Activity (% Inhibition)	Ferric Reducing Power Assay (µmol Fe (II)/g DW)
Harvesting age (HA)
Week 8	70.40 ± 0.36 ^c^	467.41 ± 10.25 ^c^
Week 12	72.31 ± 0.40 ^b^	497.64 ± 4.50 ^b^
Week 16	74.04 ± 0.30 ^a^	518.88 ± 8.79 ^a^
Harvesting frequency (HF)
Harvest 1	74.76 ± 0.38 ^a^	537.72 ± 8.79 ^a^
Harvest 2	72.33 ± 0.34 ^b^	490.37 ± 6.44 ^b^
Harvest 3	69.67 ± 0.34 ^c^	455.84 ± 8.31 ^c^
F test		
Harvesting age	**	**
Harvesting frequency	**	**
HA × HF	ns	*

Data are means of five replication measurements ± standard deviation. Means with different letters (a, b, c) at different levels of harvesting age and harvesting frequency were significantly different at *p* < 0.05 based on Duncan Multiple Range Test (DMRT) mean separation. DW: dry weight. * and ** are significant at *p* < 0.05 and *p* < 0.01, respectively. ‘ns’ is not significant.

**Table 3 molecules-25-02833-t003:** Quantification of bioactive compounds of *C. nutans* at varying levels of harvesting age and harvesting frequency.

Treatment	C-glycosyl Flavone (µg/g)				
	Shaftoside	Iso-Orientin	Orientin	Iso-Vitexin	Vitexin
Harvesting age (HA)					
Week 8	5.32 ± 0.08 ^a^	1.59 ± 0.05 ^a^	0.56 ± 0.31 ^a^	0.45 ± 0.36 ^a^	0.09 ± 0.26 ^a^
Week 12	7.12 ± 0.55 ^a^	1.31 ± 0.96 ^a^	0.65 ± 0.11 ^a^	0.29 ± 0.22 ^a^	0.33 ± 0.52 ^a^
Week 16	7.39 ± 0.91 ^a^	2.31 ± 0.68 ^a^	0.66 ± 0.36 ^a^	0.65 ± 0.82 ^a^	0.32 ± 0.81 ^a^
Harvesting frequency (HF)					
Harvest 1	11.27 ± 0.91 ^a^	3.07 ± 0.84 ^a^	1.09 ± 0.28 ^a^	0.83 ± 0.88 ^a^	nd
Harvest 2	5.37 ± 0.29 ^b^	0.78 ± 0.31 ^b^	0.56 ± 0.41 ^ab^	0.21 ± 0.14 ^a^	0.51 ± 0.89 ^a^
Harvest 3	3.20 ± 0.94 ^b^	0.49 ± 0.10 ^b^	0.23 ± 0.08 ^b^	0.36 ± 0.37 ^a^	0.23 ± 0.26 ^ab^
F test					
Harvesting age	ns	ns	ns	ns	ns
Harvesting frequency	**	*	**	ns	ns
HA × HF	ns	ns	ns	ns	ns

Data are means of five replication measurements ± standard deviation. Means with different letters (a, b, c) at different levels of harvesting age and harvesting frequency were significantly different at *p* < 0.05 based on Duncan Multiple Range Test (DMRT) mean separation. DW: dry weight. * and ** are significant at *p* < 0.05 and *p* < 0.01, respectively. ‘ns’ is not significant.

**Table 4 molecules-25-02833-t004:** Pearson correlation of herbal yield, phytochemicals, antioxidants, and bioactive compounds for the harvesting age of *C. nutans.*

	Variables	1	2	3	4	5	6	7	8	9	10
1	Yield	1									
2	TPC	0.94 **	1								
3	TFC	0.90 **	0.74 *	1							
4	FRAP	0.83 *	0.82 *	0.88 *	1						
5	DPPH	0.91 **	0.97 **	0.81 *	0.89 *	1					
6	Shaftoside	0.36	0.38	0.20	0.13	0.26	1				
7	Iso-orientin	0.37	0.32	0.45	0.44	0.23	0.09	1			
8	Orientin	0.25	0.25	0.28	0.48	0.27	0.28	0.03	1		
9	Iso-vitexin	0.21	0.29	0.06	0.09	0.21	0.01	0.12	0.49	1	
10	Vitexin	0.41	0.32	0.31	0.47	0.40	0.22	0.29	0.36	0.38	1

* and ** are significantly affected at *p* < 0.05 and *p* < 0.01, respectively. TPC = Total phenolic content, TFC = Total flavonoid content.

**Table 5 molecules-25-02833-t005:** Pearson correlation of herbal yield, phytochemicals, antioxidants, and bioactive compounds for the harvesting frequency of *C. nutans.*

	Variables	1	2	3	4	5	6	7	8	9	10
1	Yield	1									
2	TPC	−0.99 **	1								
3	TFC	−0.89 **	0.89 *	1							
4	FRAP	−0.93 **	0.97 **	0.90 *	1						
5	DPPH	−0.98 **	0.99 **	0.91 *	0.97 **	1					
6	Shaftoside	−0.95 **	0.92 *	0.69 *	0.82 *	0.91 *	1				
7	Iso-orientin	−0.84 *	0.85 *	0.60	0.84 *	0.84 *	0.86 *	1			
8	Orientin	−0.91 **	0.92 *	0.74 *	0.91 *	0.90 *	0.90 *	0.88 *	1		
9	Iso-vitexin	−0.43	0.42	0.40	0.42	0.42	0.25	0.15	0.19	1	
10	Vitexin	−0.35	−0.37	−0.14	−0.46	−0.32	−0.26	−0.46	−0.40	−0.27	1

* and ** are significantly affected at *p* < 0.05 and *p* < 0.01, respectively.

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
