# Peer review of "Alterations in Herbage Yield, Antioxidant Activities, Phytochemical Contents, and Bioactive Compounds of Sabah Snake Grass (Clinacanthus Nutans L.) with Regards to Harvesting Age and Harvesting Frequency"

_molecules, 2020, doi:10.3390/molecules25122833_

Round 1
Reviewer 1 Report
The manuscript is interesting. Please attend the following comments before aceptance:
“Due to the presence of this compound, 49 medicinal plants have been utilized extensively in the world of nutraceuticals as an alternative 50 regiment against various diseases.” Please don’t confuse the terminology of nutraceutical, medicinal plants and herbs. Definitions can be found in the following manuscript: https://www.sciencedirect.com/science/article/pii/S0924224418308616
It is important to include in the discussion a section about the correlation between the antioxidant activity and the compounds identified, authors can use as basis the following manuscript: https://onlinelibrary.wiley.com/doi/full/10.1111/j.1750-3841.2009.01352.x
What is the disadvantage of using the Folin-Cioucalteu assay for determining the total phenolic content, did you consider when analyzing the results: https://pubs.rsc.org/en/content/articlelanding/2013/ay/c3ay41125g/unauth#!divAbstract
Author Response
Point 1: “Due to the presence of this compound, 49 medicinal plants have been utilized extensively in the world of nutraceuticals as an alternative 50 regiment against various diseases.” Please don’t confuse the terminology of nutraceutical, medicinal plants and herbs. Definitions can be found in the following manuscript: https://www.sciencedirect.com/science/article/pii/S0924224418308616
Response 1: Terminology nutraceuticals has been change to pharmaceuticals. Word pharmaceutical best describe compounds relating medicinal drugs (Page 2, line 49)
Point 2: It is important to include in the discussion a section about the correlation between the antioxidant activity and the compounds identified, authors can use as basis the following manuscript: https://onlinelibrary.wiley.com/doi/full/10.1111/j.1750-3841.2009.01352.x
Response 2: Correlation between antioxidants and C-glycosyl flavone has been included in the revised articles. However, there is quite lacking on literatures to compare with present analysis on the specific compounds. Hence, the discussions are made with general facts and knowledge related to the compounds (Page 7, line 230-245)
Point 3: What is the disadvantage of using the Folin-Cioucalteu assay for determining the total phenolic content, did you consider when analyzing the results: https://pubs.rsc.org/en/content/articlelanding/2013/ay/c3ay41125g/unauth#!divAbstract
Response 2: For this study, we are not really focusing on the effect of certain methods used on TPC of samples. Method is chosen based on its convenient procedures which essential for completing the objective of this study. However, the suggestion would be highlighted for further analysis in next experiment
Other corrections and alterations on the revised article
- Are the results clearly presented?
- The results on antioxidant activity (radical scavenging activity and ferric reducing power assay) has been improved with figure and discussion correlation analysis between two assays. (Page 4-5, line 144-165).
- Are the conclusions supported by the results?
- The conclusions have been improved to support edited results (Page 10, line 359-369)
- References has been altered and edited (Page 11-14, line 389-587)
Reviewer 2 Report
The manuscript reported the identification of active compounds in Sabah snake grass or Clinacanthus nutans. Although the results are interesting, what makes this plant product beneficial to health needs to be elaborated with more experimental data. It is obvious that the chemical contents of a plant vary with growth. What are the importance of findings related to growth? Also the identification of plant compounds cannot be determined by the HPLC alone. Finally, more updated references on the plant and the related compounds need to be cited.
Author Response
1. The manuscript reported the identification of active compounds in Sabah snake grass or Clinacanthus nutans. Although the results are interesting, what makes this plant product beneficial to health needs to be elaborated with more experimental data. It is obvious that the chemical contents of a plant vary with growth. What are the importance of findings related to growth?
- The importance of the study is added in the introduction and conclusion. It is true that chemical contents of plant vary with growth but in this experiment emphasized more on how chemicals differ with the interaction effect between growth and harvesting frequency. This is because most of the farmers in Malaysia to be specific, often harvest in a relatively high number of times and neglect the fact that quality is reduced. Therefore, this experiment was done to provide that information that would later on benefits the farmers and the researchers in the future. [Page 2 (line 74-77), page 10 (line 359-369)].
2. Also the identification of plant compounds cannot be determined by the HPLC alone.
- It is true that the results related to chemical compounds are not suffice to only just by HPLC analysis alone. We do need to add more data but because this experiment was conducted last 2 years, the samples are no longer available to conduct further analysis. To repeat the experiment would take time (a year to be completed in field study alone) to collect the samples and it is impossible to be done for a short period of time. However, the comments and suggestions will be highlighted for the next experiment of this specific plant.
3. Finally, more updated references on the plant and the related compounds need to be cited.
- The references are updated in the introduction, results and discussion and references (Page 2 (line 55-59), page 7 (line 231-245), page 11-14).
Round 2
Reviewer 1 Report
The authors attended the comments from the Reviewer.